# Casting a Wider Net: Differentiating between Inner Nuclear Envelope and Outer Nuclear Envelope Transmembrane Proteins

**DOI:** 10.3390/ijms20215248

**Published:** 2019-10-23

**Authors:** Mark Tingey, Krishna C. Mudumbi, Eric C. Schirmer, Weidong Yang

**Affiliations:** 1Department of Biology, Temple University, Philadelphia, PA 19121, USA; Mark.Tingey@temple.edu; 2Department of Pharmacology, Yale University School of Medicine, New Haven, CT 06510, USA; Krishna.Mudumbi@yale.edu; 3Yale Cancer Biology Institute, Yale University, New Haven, CT 06516, USA; 4Wellcome Centre for Cell Biology, University of Edinburgh, Edinburgh EH8 3BF, UK; E.Schirmer@ed.ac.uk

**Keywords:** NETs, inner nuclear membrane, outer nuclear membrane, nuclear envelope

## Abstract

The nuclear envelope (NE) surrounds the nucleus with a double membrane in eukaryotic cells. The double membranes are embedded with proteins that are synthesized on the endoplasmic reticulum and often destined specifically for either the outer nuclear membrane (ONM) or the inner nuclear membrane (INM). These nuclear envelope transmembrane proteins (NETs) play important roles in cellular function and participate in transcription, epigenetics, splicing, DNA replication, genome architecture, nuclear structure, nuclear stability, nuclear organization, and nuclear positioning. These vital functions are dependent upon both the correct localization and relative concentrations of NETs on the appropriate membrane of the NE. It is, therefore, important to understand the distribution and abundance of NETs on the NE. This review will evaluate the current tools and methodologies available to address this important topic.

## 1. Introduction

The eukaryotic nuclear membrane consists of two separate lipid bilayers, the inner nuclear membrane (INM) and the outer nuclear membrane (ONM) that are separated by a perinuclear space of approximately 30–50 nm [1,2]. Both INM and ONM contain unique sets of nuclear envelope transmembrane proteins (NETs) that must target to their respective compartments after synthesis in the endoplasmic reticulum (ER). The ONM is contiguous with the ER and fuses with the INM at sites where nuclear pore complexes (NPCs) are inserted, often called the pore membrane [3]. NPCs are megadalton complexes built of more than 30 nucleoporin proteins that regulate directed transport of proteins and RNA in and out of the nucleus through their central channel. Along with the NE, the NPC provides a barrier against the free diffusion of large molecules into the nucleus. In the NPC, this barrier in the central channel with a narrowest waist of ~50 nm is formed by intrinsically disordered phenylalanine-glycine (FG) motifs on one thirds of nucleoporins [4,5]. In addition to the central channel, NPCs could also have ~10-nm peripheral channels between their core protein mass and the membrane [6,7]. Though these channels are not well characterized, it has been proposed that these peripheral channels facilitate the transit of INM NETs to their functional sites on the inner face of the NE. 

It is critical for the cell to allow INM NETs into the nucleus while excluding ONM NETs from entering as many NETs have important functions on their designated membrane. These functions can range from cell and nuclear migration to connecting the rest of the cell to the genome to regulating genome functions. Both INM and ONM NETs provide structure to their respective membranes, for example, many INM NETs bind the lamina intermediate filament network underlying the INM while the ONM contains NETs that connect to all three major cytoplasmic filament systems [8,9,10,11,12,13,14,15]. There are also connections across the lumen of the NE between INM and ONM NETs that form the linker of nucleoskeleton and cytoskeleton (LINC) complex, which is central to mechanosignal transduction regulating the genome, cell and nuclear mechanical stability, and providing mechanical connections needed for cell and nuclear migration [16,17]. Accordingly, many INM NETs interact with DNA, chromatin proteins, chromatin-remodeling enzymes, transcription factors, transcriptional repressors, and even splicing factors [18,19,20,21,22,23]. INM proteins also contribute significantly to 3D spatial genome organization, which is a major factor in fine-tuned regulation of the genome during tissue development. Disruption of this complex interactome can result in a number of pathological conditions, often termed as nuclear envelopathies or laminopathies, most of which are highly tissue-specific, thus underscoring the importance of INM protein function in development [24,25,26].

In order to better understand the complex interactome of the nuclear envelope and these diseases, it is becoming increasingly apparent that differentiating between INM or ONM position is a critical question. At the simplest level, understanding the function of a NET requires also knowing whether it connects the NE to the cytoplasm or to the genome, thus knowing the INM/ONM distribution answers this question. This distribution is generally not absolutely binary because NETs are first synthesized in the ER and so will never be 100% in the INM, especially if NETs freely diffuse in both directions between the ONM and INM until they find a binding partner. Thus, a very high INM:ONM ratio likely indicates a more directed mechanism for translocation that might be receptor mediated, similar to NLS-mediated directional transport of soluble proteins through the central channels of the NPCs. This ratio information can also be important to gain insights about functioning of different pools of NETs because several NETs have multiple cellular locations and being able to isolate and distinguish if they perform different functions in these locations requires understanding also what controls their targeting. For example, a subunit of the plasma membrane Na,K-ATPase was separately shown by immunogold electron microscopy to have a non-mitochondrial pool in the inner nuclear membrane and this pool functions as a co-regulator of transcription [27]. Particularly in these cases more information about the INM:ONM ratio and translocation rates can help direct research efforts towards different types of mechanisms ranging from post-translational modifications that might create a cryptic and novel transport sequence to different splice variants with different targeting sequences. This latter is particularly relevant considering that NETs such as Lap2 have at least half a dozen splice variants that have never been carefully compared for their INM:ONM ratios or translocation and a recent study indicated that NETs in general have more splice variants and particularly more tissue-specific splice variants than non-NE proteins [28]. It is noteworthy that the proteins with tissue-specific splice variants includes both transmembrane and non-transmembrane nucleoporins that make up the core structure of the NPC. 

Due to the close proximity of the ONM and INM, electron microscopy (EM) remains the only unequivocal method of determining a NET’s location. However, this only provides a snapshot of potential NET locations and, by extension, their potential binding partners and functions. This exposes a critical need for assays capable of distinguishing between INM and ONM proteins accurately in vivo. Therefore, to truly better understand the involvement of NETs in cellular functions, it is critical to develop new methods that are capable of distinguishing the localization of NETs between INM and ONM with spatial and temporal accuracy. In recent years, many existing technologies have been repurposed to interrogate this question. However, many of these technologies suffer from their own shortcomings. This is true for imaging techniques, biochemical methods, and bioinformatic approaches. Within this review, we evaluate several cutting-edge technologies within the context of interrogating the location of NETs on the nuclear envelope. 

## 2. Determining NET Location

### 2.1. Determining the Spatial Location of NETs on the NE

Determining the membrane distribution of NETs remains a difficult question to answer due to the close proximity of the INM and ONM as well as the transient nature of many NETs. An important caveat to all the research on this question is that most work relies upon exogenously expressed tagged proteins and one of the core hypotheses to how NETs get to the INM is that they diffuse laterally in the membrane and remain in the INM by binding lamins or chromatin. If correct, this “lateral-diffusion retention” hypothesis would limit binding sites in the INM to the levels of the binding partner; thus, exogenous overexpression of a NET that had limiting amounts of the INM binding partner would result in large pools of the NET moving freely in the INM, ONM, and ER. At the same time, many NETs have a plethora of splice variants. Lap2β has at least 6 different splice variants [29] and many tissue-specific variants of NETs that have not yet been cloned or characterized are indicated [28]. Correspondingly, staining with antibodies cannot distinguish splice variants. Due to the complexity and limitations of this problem, a variety of technologies have been repurposed or developed to evaluate a NET’s localization. 

#### 2.1.1. Electron Microscopy

Often considered the gold standard for determining the localization of NETs, immunogold-label electron microscopy makes use of small particles of gold bound to an antibody to generate extremely high-resolution images. Where the gold nanoparticle is present, a dark spot will be present on the image. While small, ~1 nm, particles of gold are possible to generate and label with, it becomes difficult to differentiate between organic material and the gold label [30]. Furthermore, due to the primary and secondary antibody labeling method, the gold particle will typically be 15–30 nm away from the molecule of interest, which can cause issues with distinguishing between ONM and INM localizations [30]. The potentially large distance between the gold particle and the antigen when using primary and secondary antibodies can be circumvented through the use of gold-labeled nanobodies [31,32]. Finally, due to the high density of the gold particle it will still be the strongest signal even when not in the same plane as the stained membrane; thus sectioning may cut the NE at an angle that makes it appear that a gold particle on one side of the membrane is actually on the other. Hence, a reasonable percentage of particles might appear to be in the lumen despite that the region of the protein being labeled is 100% cytoplasmic or nucleoplasmic and luminal-appearing particles cannot be determined for INM or ONM localization. 

With this difficulty in mind, the diameter of the gold nanoparticle is of critical importance. Until the development of scanning electron microscopy (SEM) instruments equipped with field emission guns, the limited resolution of SEM required the use of gold nanoparticles larger than 15 nm. However, modern SEM allows for resolution similar to that of transmission electron microscopy (TEM), approximately 0.5 nm [33]. 

The extreme localization precisions achievable through immunogold-label electron microscopy has been used to provide direct evidence of the localization of a small subset of NETs [34,35,36]. However, this approach is ponderous, expensive, and only viable in fixed cells; thereby limiting the efficacy and feasibility of this approach. In addition, the aforementioned requirement that the samples be fixed, limits the viability of this technique as it pertains to translocation of NETs.

#### 2.1.2. Differential Membrane Permeabilization

Historically it was thought that, as most INM proteins associate with the nuclear lamina, the resistance of a protein to a pre-fixation detergent extraction indicated INM localization. It has since been shown that several ONM proteins interact with cytoskeletal filaments and so this is no longer a clear determination of INM localization; however, a variant involving differential detergent extraction can still provide information on INM/ONM localization. This method takes advantage of the fact that digitonin preferentially pokes holes in membranes containing cholesterol. While the cell membrane contains large amounts of cholesterol, the ER and its contiguous ONM appear to contain very little cholesterol [37]. Thus, one can fix cells and then preferentially permeabilize the plasma membrane with digitonin and stain with antibodies [38]. Proteins on the ONM will stain strongly, but the nuclear membrane will prevent the antibodies accessing the INM thereby preventing staining [39,40]. Some proteins will have partial pools in both the INM and ONM, therefore a separate staining of cells permeabilized with Triton X-100 will show the full staining with the antibodies. If this stain shows a difference in the relative staining patterns, it can be inferred that there are pools in both membranes. Similarly, if Green Fluorescence Protein (GFP)-tagged proteins are exogenously expressed, the relative GFP signal intensity in different membranes compared to the antibody staining can be compared to determine at least a population in the INM [41,42]. This approach can also be used to determine membrane topology if mapped antibodies are used. This is because the luminal domains in both the ER and nuclear envelope will be protected in the digitonin-permeabilized cells. These approaches are still used, but trap and super resolution approaches are more in favor due to the difficulties involved in titrating the proper amount of digitonin to use, as it is possible with high levels, or lengths of digitonin treatment, to also poke holes in the nuclear membrane.

#### 2.1.3. Rapamycin Trapping

One rather clever approach to surmounting diffraction limitations is the rapamycin trap. This method is based upon the forced protein dimerization technique first reported by Chen et al. [43] which exploits the binding affinity of rapamycin. Rapamycin binds to both the 12 kDa FK506 binding protein (FKBP12) and the 100 amino acid domain of the mammalian target of rapamycin (mTOR) protein, also known as FKBP-rapamycin binding domain (FRB). This system has been used to tackle the challenge of NET localization by engineering two chimeric proteins: (1) NETs of interest expressing an FRB domain and a fluorescent tag. (2) A “trap” protein limited to the nucleus consisting of a complimentary nuclear localization signal (NLS) bearing glutathione s-transferase (GST) sequence tagged with an FKBP12 sequence and a different fluorophore from the aforementioned NET chimera (Figure 1a). In the absence of rapamycin, fluorescence microscopy will show the fluorescently-tagged NET in a ring around the periphery of the nucleus whether it is in the INM or the ONM while the soluble FKBP12-tagged GST-NLS will diffuse equally throughout the nucleus (Figure 1b). Upon rapamycin treatment, the membrane bound FRB-tagged NET will associate with rapamycin, which in turn will recruit the soluble FKBP12-tagged protein only if the NET is in the INM. Such an association will result in a distinct redistribution of the fluorophore-tagged GST-NLS protein to a ring-like staining at the nuclear periphery that can be visualized by fluorescence microscopy (Figure 1c). In contrast, if the NET is restricted to the ONM the nuclear-restricted FKBP12-tagged protein will not move to the nuclear periphery, but remain diffuse in the nucleoplasm [44,45]. This method provides a very clear condition, which when met, provides very strong support for the protein being present on the INM. However, this methodology only evaluates the presence of a protein on the INM and does not distinguish whether a separate pool can reside simultaneously in the ONM. Furthermore, the addition of two tags, the fluorophore and the FKBP12 sequence, introduces a greater possibility of error in this system, allowing for the possibility that in vivo wild-type interactions may be significantly different from what is observed. Nonetheless, there are many strengths to this system, including that it can be adapted in many ways to address related questions. For example, if the nuclear trap protein has a lamin-binding site that keeps it stably at the INM, then a NET that freely diffuses with different subcellular pools can be followed for its dynamics until it gets trapped at the INM after rapamycin is added. 

This approach has been used to test requirements for targeting a heterologous reporter to the INM in a study supporting the lateral-diffusion retention hypothesis that found an ATP requirement for translocation of the reporter to the INM [46]. However, it was also used in a study supporting the lateral-diffusion retention hypothesis that argued against an ATP requirement for translocation [47]. 

#### 2.1.4. Split GFP

Another proximity-based interaction system utilizes Superfolder GFP, which is capable of being split asymmetrically into two parts, GFP_11_ (3 kD) and GFP_1–10_ (24 kD). Individually, these two constructs do not fluoresce. However, they can be reconstituted into a fluorescing GFP (27 kD) when the two pieces are expressed within the same cellular compartment and associate (Figure 2a) [48,49]. This assay was adapted to identify the localization of NETs on the NE of *Saccharomyces cerevisiae* by Smoyer and colleagues [50]. To accomplish this, they created reporter proteins by fusing a yeast-codon optimized GFP_11_ and mCherry to a nuclear reporter (GFP_11_-mCherry-Pus1), ONM/ER surface reporter (GFP_11_-mCherry-Scs2TM), ER lumen reporter (mCherry-Scs2TM-GFP_11_), and a cytoplasmic reporter (GFP_11_-mCherry-Hxk1) (Figure 2e). NETs were then selected and fused with the complimentary GFP_1–10_ fragment. Each NET-GFP_1–10_ construct was then expressed with each of the reporters individually and observed in the green fluorescence channel. Since the location of the reporters is well established, if the nuclear protein fused to GFP_1–10_, was present in the same compartment as the reporter protein, fluorescence in the green channel would be detected, and thus, the localization of the NET in question could be determined (Figure 2c,d). To confirm the accuracy of the system, control proteins were generated which localized to specific regions, thereby confirming the location of the experimental constructs (Figure 2e). Split GFP is a powerful and elegant system for qualitatively determining if a NET localizes to the INM. However, the system does not account for NETs with a dual role that may be present on both the INM and ONM as it does not allow for derivation of information about ONM quantity and proportion to the INM. Furthermore, similar to the rapamycin trap, split GFP carries with it the potential that adding several tags may alter the behavior of NETs, which may lead investigators to reach erroneous conclusions. Thus, despite being an elegant and straightforward approach to verifying the presence of NETs on the INM, the split GFP system is limited in its capacity to provide further information about the translocation and proportion of INM NETs.

#### 2.1.5. MIET

Recently, Metal-Induced Energy Transfer (MIET), a technique that relies upon the principals of non-radiative electromagnetic energy transfer, was used to probe the distribution of NETs on the ONM and INM [51]. This method is similar to the more commonly used technique of Förster resonance energy transfer and fluorescence lifetime imaging microscopy (FRET/FLIM). Here, donor fluorophores close to a metal surface interact with surface plasmons and transfer their energy to the metal thereby reducing their fluorescence lifetime (τ) in a direct relationship with their distance from the metal surface. This technique can work over the range of about 150 nm, and can therefore help determine the location of proteins on the part of the NE that is close to the bottom of a cell and near the metal surface [51,52].

This technique has recently been utilized to generate a topography of the nuclear envelope. This was done by tagging landmark proteins Lap2β on the INM, and NUP358 for the ONM and then using MIET to localize the landmark proteins. MIET boasts an impressive axial localization of 2.5 nm, which allows for a very accurate differentiation between the INM and ONM, as well as a very accurate measurement of the basal region of the perinuclear space [51]. While no specific experiments have been published using MIET to differentiate membrane location of NETs, we propose that this technology could potentially be utilized to accurately determine the membrane location of NETs. MIET is a very interesting concept with many potential applications, however it is not without limitations. The temporal resolution of this technique is currently too low to provide real-time mobility measurements of NETs. Furthermore, the lateral resolution of this technique is still diffraction limited and therefore unable to determine the distribution of NETs on the ONM and INM. However, this may be overcome by employing Single-Molecule Localization Methods (SMLM) to improve the lateral precision. 

#### 2.1.6. Ensemble FRAP

Since the initial breakthrough experiments in the 1970s, fluorescence recovery after photobleaching (FRAP) has become an essential tool to determine the mobility and diffusion of transmembrane proteins embedded in lipid bilayers [53,54,55]. Since then, FRAP has been used on many membranous structures, including the plasma membrane [56,57,58], ER [59], and NE [34]. FRAP allows one to distinguish the diffusivity of molecules on a membrane as well as the fraction of immobile molecules that are unable to diffuse due to interactions with other macromolecules. For most NETs tested, however, the immobile fraction was sufficiently high that FRAP was principally measuring the translocation of protein accumulated in the ER into the INM instead of mobility of protein within the NE. While this is straightforward on single-membrane structures such as the plasma membrane, it becomes more convoluted on the NE due to the ~40 nm distance of the ONM and INM. FRAP relies on diffraction limited imaging which does not have the resolution to distinguish between the two membranes. Therefore, any information about NET diffusion coefficients or immobile fractions inherently is an average of behavior of the specific protein in question on both of the membranes. 

#### 2.1.7. Airyscan Confocal Microscopy and Differential Labeling

The expertise required for super-resolution or electron microscopy may be untenable for many research labs, therefore, Airyscan confocal microscopy is an attractive alternative to these other techniques due to its accessibility and comparative affordability. The Airyscan confocal microscope’s principle of operation makes use of multiple extremely sensitive GaAsP detectors for a single illumination point. The detector consists of 32 detector elements arranged in a compound eye fashion. As the image is scanned, each detector records a portion of the whole. The resultant images are then concatenated and a point spread function (PSF) is generated. This PSF allows for a sub-diffraction limit image to be generated with a lateral resolution of approximately 140 nm and an axial resolution of approximately 350 nm [60,61,62]. This resolution is not sufficient to differentiate between INM and ONM. However, the addition of differential labeling overcomes this weakness by enabling investigators to create a landmark on the INM or ONM. 

Previous publications have identified landmark proteins which localize to the ONM or the INM respectively. Labeling these known protein markers with a fluorescent protein (i.e., mCherry) and NETs of interest with a different color fluorophore (i.e., eGFP) allows one to study colocalization. This approach was used to great effect by Groves et al. to better understand how NETs are targeted to the INM in plants. In this study, ER tail-anchored proteins were tagged with an NLS and GFP and compared to the localization of calnexin-mCherry, a well characterized protein located exclusively on the ER and ONM. Since the Airyscan microscopy method does not have enough resolution to visually distinguish between the ONM and INM, line scans of the NE were used to determine co-localization of the two proteins (Figure 3a,b). However, even line scans are limited by the overall resolution of the system, therefore several statistical analyses had to performed on the line scan results before providing satisfying conclusions [63].

While this approach is relatively simple and easy to perform, it unfortunately has several pitfalls that can drastically affect the results. First, overexpression of NETs can often result in their mislocalization on the NE due to the leaky nature in which proteins are regulated by the NPC [64]. This is especially detrimental if the ONM marker, calnexin, is found on the INM as the line scan will show false colocalizations. Second, there is an inherent amount of uncertainty associated with fitting a line scan to determine the peaks. The resultant error makes using these values to colocalize proteins precarious. Line scans are a one-dimensional analysis method and do not provide information regarding diffusion or relative enrichment on the ONM or INM.

#### 2.1.8. Super-Resolution Microscopy

As was discussed previously, the diffraction limit of light microscopy is approximately 250 nm. This limit can be overcome by using super-resolution (SR) microscopy techniques, which are theoretically capable of providing an image resolution between 100 to 20 nm. So far, three classes of SR techniques have been applied to imaging NETs: Structured illumination microscopy (SIM), stimulated emission depletion (STED) microscopy, and single-molecule localization microscopy (SMLM). While all three of these techniques can break the diffraction barrier of light, they vary drastically in approach. SIM relies on a diffraction pattern or grating placed in front of the excitation laser beam. These patterns are then moved and rotated several times to produce a Moiré effect allowing one to discern high frequency signals relating to fine cellular structures, reaching resolutions of ~100 nm laterally and ~250 nm axially. This technique was originally used to image the localization of the nuclear lamina and the INM protein Lap2β in relation to either an NPC protein of the nuclear basket that protrudes into the nucleoplasm by ~50 nm or an NPC protein of the cytoplasmic filaments that similarly protrude into the cytoplasm [65]. Thus, the INM or ONM localization was determined based on whether a protein was closer to one or the other NPC protein and the NPC proteins were in fact separated by well over 100 nm distance. This approach was subsequently expanded to study the localization and interactions between INM proteins, ONM proteins, and the cytoskeleton [66], as well as to systematically analyze the localization of 21 novel NETs identified by proteomics [41,67,68]. While this technique can be performed on live cells, the need to take thousands of images makes the image acquisition rate far too slow to detect the fast dynamics of transmembrane protein diffusion on the NE. It is important to remember when applying this technique that the NPC proteins used as landmarks are penetrating into the nucleoplasm or cytoplasm by ~50 nm. Thus, if the protein being interrogated has its tag or epitope being recognized by an antibody in the lumen or near the membrane on the nucleoplasmic side as opposed to similarly penetrating into the nucleoplasm, a clear distinction on its localization may not be possible. 

Improving significantly upon the resolution of SIM is STED microscopy, which uses a depletion laser that is overlapped with an excitation laser. This depletion laser depletes the excited state of fluorophores on the outer edges of the excitation laser targets so that only the inner most region of the excitation laser will excite fluorophores to emit photons [69]. In other words, this method reduces the effective PSF of the excitation laser below the diffraction limit of light, giving lateral resolution of ~50 nm and an axial resolution of ~150 nm in 3D STED applications [70]. Using this technique in combination with FRAP, Giannios and colleagues examined the localization and mobility of lamin B receptor (LBR) and concluded that the mobility of LBR is greatly affected by the interfaces between ER tubules and the ONM as well as discrete LBR microdomains. Mobility was primarily determined by using FRAP/FLIP techniques, whereas localization was determined by fixing the cells and performing STED, which, naturally, does not preserve the dynamics of a live cell system.

Finally, SMLM takes advantage of the photophysical properties of fluorescent probes and relies on the blinking or ‘on/off’ switching of the probes. The two most commonly SMLM methods are stochastic optical reconstruction microscopy (STORM) and photoactivated localization microscopy (PALM), both which rely on the blinking of fluorophores, but differ in that STORM relies on the blinking of organic dyes, whereas PALM uses the photoactivation or photoswitching of genetically modified fluorescent proteins. These blinking events spatially and temporally separate fluorophores, allowing each active emitter to be distinctly observed and localized with mathematical functions to find the centroid. Thousands of subsets of active emitters are imaged and then reconstructed to recreate the original image. This powerful technique has a very high localization precision (~20 nm laterally and ~50 axially), however, it is time consuming and not ideal for capturing the dynamic movement of NETs on the NE. As such this methodology has mostly been used to localize different lamin subtypes within the nuclear lamina in fixed cells [71,72]. It is noteworthy, however, that application of multiple of the above-listed SR approaches to this problem of lamin subtype localization yielded somewhat differing results.

Recently, however, a technique was developed harnessing both SMLM and FRAP—named single-molecule fluorescence recovery after photobleaching (smFRAP)—to try and capture the dynamic movement and distribution of NETs on both the ONM and INM [73]. By photobleaching a small spot on the NE, the local concentration of fluorescently functional EGFP tagged NETs was brought to near zero, and the recovery of new, fluorescently functional, molecules on the ONM or INM was recorded using SMLM. Photobleaching and observing only a small spot at the equator of the NE (~0.5 µm), the ONM and INM could be treated as two parallel membranes with no overlap due to curvature. Then, by exploiting the high lateral localization precision of SMLM, the recorded events on either of the two membranes could be easily separated to determine the distribution of NETs on the respective membranes. Furthermore, observing such a small location allows for the use of a fast frame rate, preserving the natural dynamics of a live system while having the fast acquisition speed to provide information about translocation rates for NETs. However, a drawback of this system is that only a small region is imaged and used to represent the entire NE, which is not an isotropic structure. To overcome this, results measured from multiple NEs have been averaged to represent the final outcomes [66].

### 2.2. Determining the Translocation Rate of NETs

The membrane distribution of NETs can sometimes provide insight into the deeper question of how NETs import or export through the NPC and their translocation rate. While several of the aforementioned technologies are capable of providing a qualitative relative ratio of proteins on the ONM to proteins on the INM and many early studies did very elegant work to gain insights into the routes and mechanisms of translocation, in vivo direct measurement of translocation rate of NETs still remains both desirable and challengeable. However, some, when applied creatively and with mathematical modeling, have given insights into this question.

#### 2.2.1. Mathematical Modeling of NETs Translocation Rate

One of the first studies to try to get at this question creatively used the rapamycin trap described above. In this case the fusion construct carrying the FKBP was fused to the lamin binding domain of Lap2β and, as this would stay at the nuclear periphery, it was used as a trap to capture freely diffusing FRB fused to pyruvate kinase, the Lap2β transmembrane domain, and GFP. Thus, when rapamycin was added the GFP signal would begin to accumulate at the NE and the rate of this accumulation could be inferred to reflect the translocation rate [46]. However, this system still would have a background from the pool of the FRB construct already in the INM, but not yet trapped, and that in the ONM. This was improved upon slightly in a subsequent study that used a combination of data from FRAP, photoactivatable GFP, and immunogold EM to model translocation and estimate translocation rates for a variety of NETs with different characteristics in terms of length of the nucleoplasmic region, number of transmembrane spans and the presence or absence of an NLS [36]. One of the particularly interesting findings in this paper was that it indicated that earlier FRAP studies that purported to estimate mobility in the NE were most likely mostly measuring translocation. They found that photoactivation of GFP-NETs in the ER resulted in GFP signal accumulating in the NE with kinetics similar to FRAP studies while photoactivation of these GFP-NETs in the NE resulted in much slower mobility, thus indicating that most of the protein observed at the NE for these particular NETs was immobile and therefore likely tethered in the INM. Further using data from all three experimental approaches they were able to model the translocation and infer the percent immobile fraction and translocation rates of NETs NET55, Lap2β, and Lap1-L, respectively at 50–70 s, 60–80 s, and 70–140 s. 

Another study attempted to determine a translocation coefficient for just the lamin B receptor (LBR) using a slightly different approach. LBR is a well characterized INM membrane bound protein, with domains binding the B-type lamins, as well as Histones H3 and H4 and heterochromatin protein 1 (HP1). Thus it has many nucleoplasmic tethering sites consistent with the lateral-diffusion retention model of translocation and, in fact, it was the first NET studied by Soullam and Worman when presenting this hypothesis [74,75]. In addition, LBR contains 3 NLS sequences, consistent with the receptor-mediated model of translocation [76]. LBR is also a good substrate to test because, having 8 transmembrane spans, it is unlikely to be sustainable outside the membrane while it is possible that C-terminal anchored single-span transmembrane proteins never get inserted into the membrane. This study tried to determine the kinetics of LBR protein translocation using inducible expression of the LBR reporter fused to another protein making it too long to translocate through the peripheral NPC channels [77]. This reporter also had a protease site between the LBR and fusion so that before the addition of the protease, the LBR construct would be constrained to the ER/ONM. However, upon the addition of a protease, the fusion protein would be cleaved from the LBR, thus allowing it to freely localize to the INM, which is then observed over time to determine translocation kinetics and requirements. Using the data accumulated from the inducible cleavage reporter and FRAP, they were able to estimate the binding time of LBR to its nuclear binding partners as 0 to 4 min. Furthermore, they used the surface area of the NE and ER, the degradation of proteins on the ER/ONM and INM, as well as the kinetics of the inducible cleavage to determine the kinetics of LBR import. Finally, it was estimated that diffusion from the ER to the NE occurs in 5 to 15 s, and it was assumed that LBR was at a concentration of 1 µM (2.6 × 10^6^ molecule). Using these parameters, Boni and colleagues calculated the translocation rate of LBR to be 4.6 molecules per minute per NPC.

#### 2.2.2. Experimental determination of NETs Translocation Rate

A recent study by Mudumbi and colleagues used a new approach, combining smFRAP and ensemble FRAP, to directly determine the translocation rate of LBR. Using smFRAP, they were able to clearly distinguish the diffusion coefficient of LBR on the ONM and INM, along with the fraction of LBR distribution on the ONM and INM. In addition, the group used ensemble FRAP measurements of LBR to determine its mobile and immobile fractions. Finally, using previously published data about the number of LBR molecules and NPCs typically expressed in HeLa cells [78,79], they calculated the translocation rate of LBR to be approximately 5.4 molecules per minute per NPC. This method of calculating translocation rate is possible for all NETs so long as a relatively accurate quantification of total NET molecules is available. Here, the authors were able to take advantage of the high resolution of SMLM to directly determine distribution of LBR on both membranes as well as its diffusion coefficient on both membranes to calculate translocation rate, without relying on numerous theoretical assumptions.

## 3. Conclusions and Future Directions

Several microscopy techniques are able to provide the nanometer level spatial resolution required to distinguish protein localization on the ONM and INM, however they suffer from a small field of view or lack of appropriate temporal resolution. It is clear from a survey of recent literature that advances are quickly being made in the fields of microscopy and dye development that are pushing the boundaries of both temporal and spatial resolution in live cells. The application of these new technologies, together with approaches described above for measuring translocation rates from single molecules, will play an important role in addressing many outstanding questions in the area of nuclear envelope biology. For example, do different transmembrane NET ‘cargos’ have different transport kinetics? Furthermore, how might this reflect on their using distinct transport mechanisms? The application of these advances together with FRET, as shown above, and FFS (fluorescence fluctuation spectroscopy), and FCCS (fluorescence cross-correlation spectroscopy) approaches will enable further determination of which, if any, nucleoporins interact with NET cargos and if transportins might facilitate the transport process though the peripheral NPC channels. Moreover, these approaches might be also used to determine if—a totally new conceptual hypothesis—NETs could possibly themselves act as transport receptors for soluble cargos translocating through the peripheral channels of the NPCs. This could be an important backup mechanism, for example, when central channel transport is blocked by pathogens and would be consistent with the notion that the peripheral channels were actually the original mode of transport when NPCs presumably evolved from COP proteins of the ER [80] and before they acquired FGs and a directed transport mechanism. It is noteworthy that NLS deletion did not block lamin nuclear import [81] and there are actually many nuclear proteins for which an NLS has not been identified that make the possibility of multiple transport mechanisms through the peripheral channels the more likely.

One critical area is lacking on the technology side: many high-resolution biology techniques involve high laser power which causes the photobleaching of fluorophores and increases phototoxicity in the cell. This both may affect in vivo cellular dynamics and is incompatible with the need for single molecule tracking in some of the approaches discussed above. As such, there is high demand for further development of high-resolution techniques that are also gentle on the cell. This is particularly important for studying transport due to the need to simultaneously track the NET and its cargo while also following other molecules to obtain positional reference information against the NPC structure. It is also important to use these developing approaches together with CRISPR technologies to maintain endogenous expression levels and smaller tags to limit mistargeting: This is particularly important in working with the nuclear envelope because light has to pass through the whole cell to get to the nucleus and thus large amounts of out-of-focus light from inappropriate protein pools will yield poor signal-to-noise ratios. Harnessing these new methods in conjunction with the creative biological techniques presented here will help shed light in this area of nuclear biology that is poorly understood, but functionally extremely important as indicated by the many nuclear envelope-linked diseases [82,83].

One critical area is lacking on the biology side: this work will have to go hand-in-hand with developments in finding and annotating new splice variants of NETs and using these technologies that distinguish outer and inner nuclear membranes to directly compare each of these variants. One of the first NETs to be identified was Lap2, which has between 6 and 7 annotated splice variants [29,84]; however, for Lap1 which was identified at the same time, there were three bands identified by monoclonal antibodies on Western blot. Despite being discovered nearly 30 years ago, cDNAs encoding all three splice variants have yet to be annotated. A recent analysis of RNA-Seq data showed that genes encoding nuclear envelope proteins are enriched for splice variants compared to the rest of the genome as well as identifying many new tissue-specific splice variants [28]. Accordingly, it is not surprising that other studies are popping up occasionally showing important functions for novel splice variants of NETs, such as three SUN1 splice variants that each differently contribute to directional cell migration [85]. In light of the tissue-specificity of pathology in many identified nuclear envelope-linked diseases and that widely expressed proteins have mostly been implicated in these disorders [82,83], it is likely that tissue-specific splice variants that may use distinct transport pathways or exhibit distinct translocation kinetics could play a role in the disease pathomechanisms.

## Figures and Tables

**Figure 1 ijms-20-05248-f001:**
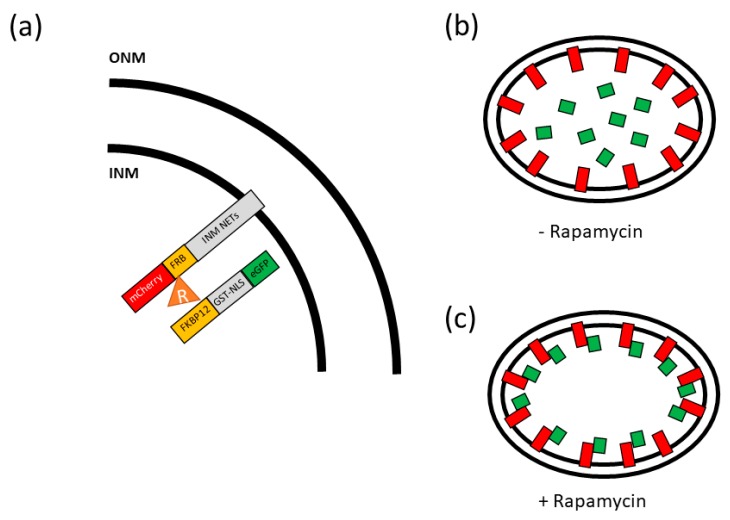
A rapamycin trap for evaluating the presence of nuclear envelope transmembrane proteins (NETs) on the INM: (**a**) A membrane bound NETs tagged with a fluorophore and FRB will dimerize with another fluorescently tagged protein with an 12 kDa FK506 binding protein (FKBP12) domain in the presence of rapamycin, represented here as a triangle. (**b**) In the absence of rapamycin, the soluble Green Fluorescent Protein (GFP) labeled protein (green blocks) will not dimerize with the membrane bound NETs (red blocks) and diffuse throughout the nucleus. (**c**) In the presence of rapamycin the soluble GFP labeled protein (green blocks) will dimerize with the membrane bound NETs (red blocks) and localize to the nuclear envelope.

**Figure 2 ijms-20-05248-f002:**
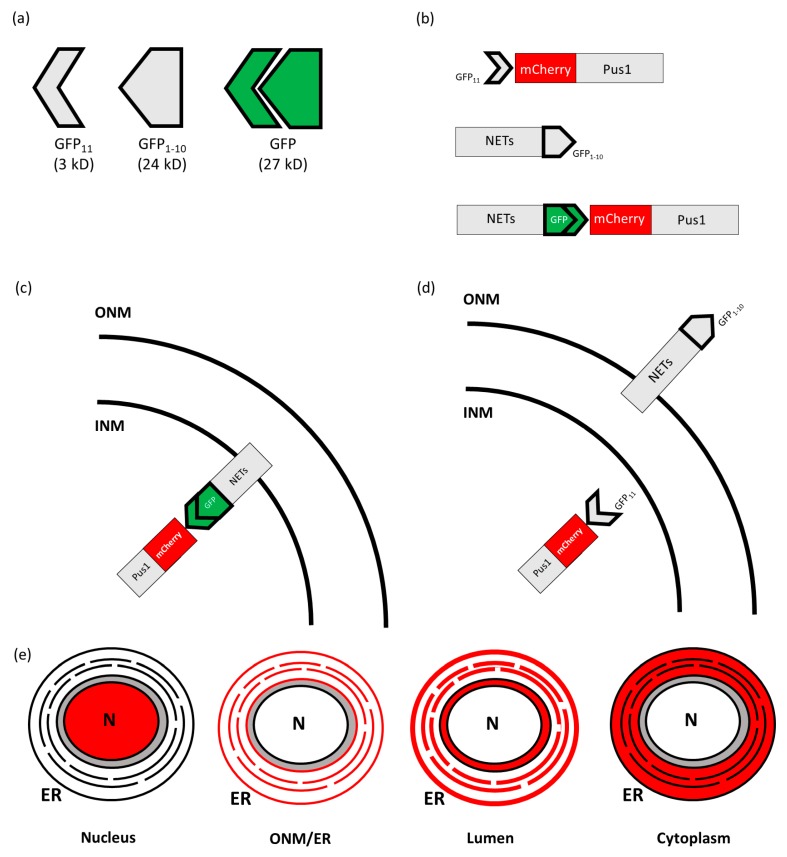
A conceptual representation of the split GFP system as it is used to identify the position of INM proteins: (**a**) Superfolder GFP can be split into two non-fluorescent components, which can also recombine into a fluorescently functional GFP. (**b**) A representation of the soluble nuclear yeast protein Pus1 tagged with mCherry and GFP_11_, a NET of interest tagged with the complimentary GFP_1–10_, the interaction between GFP_1–10_-tagged NET and the GFP_11_ reporter resulting in green fluorescence. (**c**,**d**) Representations of how the localization of NETs tagged with GFP_1–10_ and reporter proteins tagged with GFP_11_ produce green fluorescence (**c**) or fail to do so (**d**). (**e**) A representation of control proteins tagged with mCherry and GFP_11_ in the nucleus, the outer nuclear membrane (ONM) and endoplasmic reticulum (ER), the lumen, and the cytoplasm of the cell in the absence of GFP_1–10_ fused NETs of interest.

**Figure 3 ijms-20-05248-f003:**
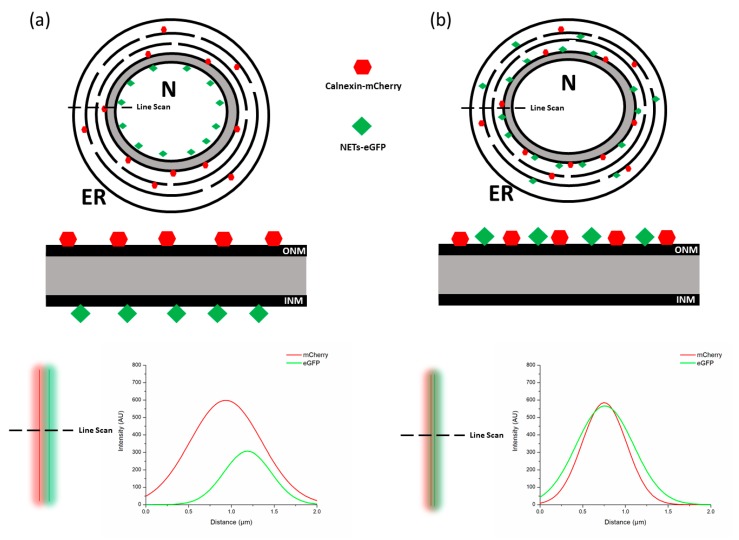
A model of differential staining using Airyscan confocal microscopy. (**a**) Calnexin tagged with mCherry (red) localizes exclusively to the ONM and ER. While the NETs of interest tagged with GFP (green) is enriched at the INM. A line scan is performed and a line profile is generated indicating that the two fluorophores do not co-localize. (**b**) A line scan of NETs tagged with GFP (green) that are enriched at the ONM and ER do co-localize with calnexin.

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
