# Peer review of "Casting a Wider Net: Differentiating between Inner Nuclear Envelope and Outer Nuclear Envelope Transmembrane Proteins"

_ijms, 2019, doi:10.3390/ijms20215248_

Round 1

Reviewer 1 Report

In this review, the authors describe techniques to spatially and temporally visualize the distribution and translocation of nuclear envelope transmembrane proteins (NETs).  They succinctly summarize each technique and sufficiently explain strengths and weaknesses.  The review is well written and covers each technique clearly highlighting the most important details.  In general, I am satisfied with this review as written.

My main suggestion is the authors expand upon their ideas in the conclusion.  After clearly laying out the strengths and weaknesses of each technique, they suggest several interesting directions forward in the conclusion.  I would find this work to be more interesting if they expanded upon these ideas and added more of their own perspective to the future of NET studies.

Author Response

In this review, the authors describe techniques to spatially and temporally visualize the distribution and translocation of nuclear envelope transmembrane proteins (NETs).  They succinctly summarize each technique and sufficiently explain strengths and weaknesses.  The review is well written and covers each technique clearly highlighting the most important details.  In general, I am satisfied with this review as written.

 My main suggestion is the authors expand upon their ideas in the conclusion.  After clearly laying out the strengths and weaknesses of each technique, they suggest several interesting directions forward in the conclusion.  I would find this work to be more interesting if they expanded upon these ideas and added more of their own perspective to the future of NET studies.

>> Thank you for your great suggestions. As shown in the revised version, we have incorporated a much more expansive conclusions section (page 11-12).

Reviewer 2 Report

In their review article Tingey and co-workers provide an excellent up-to-date overview on the methodology to study localization and dynamics of nuclear envelope trans-membrane proteins (NETs). They focus on techniques allowing us to study differential localization patterns of these proteins. The text is very easy to read and contains instructive, comprehensive figures. All main relevant methods in this context and their specific strengths and weaknesses are mentioned and described. I have only a few suggestions that may be considered by the authors in a minor revision of their nice article.

In paragraph 2.1.1 it may make sense to mention the possibility to use gold-labeled nanobodies in order to circumvent the problem of the potentially large distance between gold particles and antigen when using primary and secondary antibodies. One method to distinguish between INM vs. ONM localization of NETs is not mentioned in the paper. When isolated nuclei are stained with antibodies directed against putative INM proteins a differential treatment with and without detergent allows to distinguish between a predominant INM localization vs. localization in the ONM or both membranes.

Author Response

In their review article Tingey and co-workers provide an excellent up-to-date overview on the methodology to study localization and dynamics of nuclear envelope trans-membrane proteins (NETs). They focus on techniques allowing us to study differential localization patterns of these proteins. The text is very easy to read and contains instructive, comprehensive figures. All main relevant methods in this context and their specific strengths and weaknesses are mentioned and described. I have only a few suggestions that may be considered by the authors in a minor revision of their nice article.

In paragraph 2.1.1 it may make sense to mention the possibility to use gold-labeled nanobodies in order to circumvent the problem of the potentially large distance between gold particles and antigen when using primary and secondary antibodies.

>> Thank you for pointing this out. As suggested, we have added this important information into our revised manuscript (page 3).

One method to distinguish between INM vs. ONM localization of NETs is not mentioned in the paper. When isolated nuclei are stained with antibodies directed against putative INM proteins a differential treatment with and without detergent allows to distinguish between a predominant INM localization vs. localization in the ONM or both membranes.

>> This is great suggestion. We have added a new session “2.1.2. Differential Membrane Permeabilization” to cover the method in the revised manuscript (page 3-4).